# Analysis of Surface Energy Changes over Different Underlying Surfaces Based on MODIS Land-Use Data and Green Vegetation Fraction over the Tibetan Plateau

Jie Ma [1], Xiaohang Wen [1,*], Maoshan Li [1], Siqiong Luo [2], Xian Zhu [3], Xianyu Yang [1] and Mei Chen [1]

[1] Plateau Atmosphere and Environment Key Laboratory of Sichuan Province, College of Atmospheric Sciences, Chengdu University of Information Technology, Chengdu 610021, China; 3200101058@stu.cuit.edu.cn (J.M.); lims@cuit.edu.cn (M.L.); xyang@cuit.edu.cn (X.Y.); 2018012049@stu.cuit.edu.cn (M.C.)

[2] Key Laboratory of Land Surface Process and Climate Change in Cold and Arid Regions, Northwest Institute of Eco-Environment and Resources, Chinese Academy of Sciences, Lanzhou 730000, China; lsq@lzb.ac.cn

[3] School of Atmospheric Sciences, Sun Yat-Sen University, Zhuhai 519000, China; zhux53@mail.sysu.edu.cn

\* Correspondence: wxh@cuit.edu.cn

**Abstract:** To better predict and understand land–atmospheric interactions in the Tibetan Plateau (TP), we used Moderate Resolution Imaging Spectroradiometer (MODIS)-based land-use data and the MODIS-derived green vegetation fraction (GVF) to analyze the variation trend over the TP. The in situ observations from six flux stations ("BJ" (the BJ site of Nagqu Station of Plateau Climate and Environment), "MAWORS" (the Muztagh Ata Westerly Observation and Research Station), "NADORS" (the Ngari Desert Observation and Research Station), "NAMORS" (the Nam Co Monitoring and Research Station for Multisphere Interactions), "QOMS" (the Qomolangma Atmospheric and Environmental Observation and Research Station), and "SETORS" (the Southeast Tibet Observation and Research Station for the Alpine Environment)) at the Chinese TP Scientific Data Center were used to study the surface energy variation characteristics and energy distribution over different underlying surfaces. Finally, we used observation data to verify the applicability of the ERA-5 land reanalysis data to the TP. The results showed that the annual GVF steadily declined from the southeast parts to the northwest parts of the TP, and the vegetation coverage rate was highest from June to September. The sensible heat flux (H), latent heat flux (LE), net surface radiation ($Rn$), and four-component radiation (solar downward shortwave radiation ($Rsd$), surface upward shortwave radiation ($Rsu$), atmospheric downward longwave radiation ($Rld$), and surface upward longwave radiation ($Rlu$)) reached their maxima in summer at each station. $Rld$ did not change significantly with time; all other variables increased during the day and decreased at night. The interannual variation in H and LE shows that latent heat exchange was the dominant form of energy transfer in BJ, MAWORS, NAMORS, and SETORS. By contrast, sensible heat exchange was the main form of energy transfer in NADORS and QOMS. The Bowen ratio was generally low in summer, and some sites had a maximum in spring. The surface albedo exhibited a "U" shape, decreasing in spring and summer, and increasing in autumn and winter, and reaching the lowest value at noon. Except for SETORS, ERA-5 Land data and other flux stations had high simulation accuracy and correlation. Regional surface energy changes were mainly observed in the eastern and western parts of the TP, except for the maximum of H in spring; the maximum values of other heat fluxes were concentrated in summer.

**Keywords:** Tibetan Plateau; ERA-5 reanalysis data; surface energy; land–atmospheric interaction; different underlying surfaces

## 1. Introduction

The Tibetan Plateau (TP) is the highest plateau in the world and the largest plateau in western China, known as the "roof of the world" and "the third pole of the earth," with an

elevation between 3000 to 5000 m [1]. The plateau's high and towering terrain and complex underlying surface features significantly impact the plateau monsoon, water vapor cycle, and atmospheric vertical circulation, affecting climate change in East Asia [2–6]. In the 1970s, the first Scientific Expedition and Research on the Tibetan Plateau began. The main goal was to elucidate the history of geological development and the causes of plateau uplifting, to study the effects of uplifting on the ecological environment and human activities on local climate, and to look into the characteristics of natural conditions and resources, as well as the directions and routes for their exploitation and modification [7,8]. The Second Tibetan Plateau Scientific Expedition and Research will be based on the First Tibetan Plateau Scientific Expedition and Research, highlighting the change as the theme of investigation and research, to determine the law of change, evaluate and predict the future trend of change, and carry out ten scientific expeditions and research missions [7]. The study of the westerly monsoon synergy's evolution law, variation characteristics, and driving mechanism, as well as greater knowledge of the land–atmospheric interactions, precipitation efficiency, and the impact on the Sichuan Basin and its climate effect are all essential for revealing environmental changes on the TP [9–12]. Land–atmospheric interactions and local climate effects are the primary focus of this study. Specifically, the transfer and exchange of heat, momentum, water vapor, and carbon dioxide fluxes between the land surface and atmosphere are essential components of atmospheric interactions [13,14]. Energy and material transport are essential forcing fields for the development of convection in the atmospheric boundary layer. The thermodynamic and dynamical effects of the TP on the atmosphere are mainly influenced by the free air flow through the near-layer and boundary layer of the TP [15–18]. Land surface parameters such as green vegetation, soil texture, and soil moisture are essential factors that affect changes in surface energy flux over the TP. Moreover, owing to the wide area, complex vegetation types, and high altitude, the underlying surface characteristics significantly affect the water–energy cycle between the land surface and atmosphere. The scarce distribution of meteorological observation stations on the TP could pose a challenge to understanding the effects of the above-mentioned factors [19–21].

Ma et al. first analyzed the radiation characteristics of the period before and after the monsoon in the Nagqu area using radiation observations from the 1998 Intensification Observation Period (IOP). Observations were then compared with parameterized remote sensing results [22–24]. Li et al. found that sensible heat flux (H) is the primary energy source providing heat from the land surface to the atmosphere before the monsoon's outbreak, whereas latent heat flux (LE) is the main source of atmospheric warming during the monsoon season [25]. Studies have found that climate change in the TP exhibits a consistent warming trend at different timescales, and grasslands in semi-arid areas are highly sensitive to temperature and precipitation changes [26]. Studies have found that the H on the interannual variability of the TP shows a trend of weakening and falling at a rate of 2% per decade, with climate change and reduced wind speed over the TP identified as the causes of this phenomenon. However, the plateau's warming rate is higher than at the same latitude in eastern China, which remains unexplained [27–30]. Except for the Yarlung Zangbo River Basin, the LE was found to increase on the TP. This may be due to the increase in the net surface radiation ($Rn$) from the wetter forest cover underlying surface and the high soil moisture content caused by agricultural irrigation [31,32].

Studies have shown that diurnal variations in surface upward shortwave radiation ($Rsu$) and soil heat flux in alpine meadows are larger than those in banana plantations [33]. Net longwave radiation can affect soil-water freezing and its duration [34], the near-surface soil freeze–thaw process, heat storage, and melting of snow. Vegetation growth and non-growth periods affect surface energy non-closure [35,36]. The surface energy flux of Qomolangma has clear diurnal and seasonal variation trends that are greatly affected by the southwest monsoon. The response of the surface albedo to changes in rainfall has a lag effect. In winter, the vegetation cover in most areas of the TP is reduced, snow is present on the surface, and the surface albedo is often at the annual highest value [37,38]. Based on the analysis of the surface radiation observation data from the BJ site of Nagqu Station

of Plateau Climate and Environment, Muztagh Ata Westerly Observation and Research Station (MAWORS), Ngari Desert Observation and Research Station (NADORS), Nam Co Monitoring and Research Station for Multisphere Interactions (NAMORS), Qomolangma Atmospheric and Environmental Observation and Research Station (QOMS), and Southeast Tibet Observation and Research Station for the Alpine Environment (SETORS), it was found that the *Rsu* and surface albedo of all stations decreased on the whole. The atmospheric downward longwave radiation (*Rld*), surface upward longwave radiation (*Rlu*), net surface radiation (*Rn*), ground surface temperature, and air temperature at most observation stations showed an upward trend at the interannual scale. The amplitude of *Rlu* was more significant than that of the downward long-wave radiation. *Rn* often reaches a maximum in late spring and early summer in the Ngari area [24,39]. The variation in characteristics of the surface energy flux with time at each station has been analyzed in detail; however, the surface energy distribution has yet to be discussed further.

The above studies considered the surface energy variation characteristics of the TP in numerous ways. However, the majority of these studies focused on the TP's eastern part, with only a few addressing the western part. In addition, the majority of these studies used short-term or limited-period data, with only a few studies studying the long-term changes in land surface energy and heat fluxes. In this study, we used Normalized Difference Vegetation Index (NDVI) data from MODIS, ERA-5 Land reanalysis data, and long-term flux observation station data from six sites (BJ, MAWORS, NADORS, NAMORS, QOMS, and SETORS) in the TP of the Second Tibetan Plateau Scientific Expedition and Research to examine the long-time series variation characteristics and energy distribution differences of the surface energy fluxes on different underlying surfaces over the TP.

## 2. Data and Methods

### 2.1. Data

#### 2.1.1. Observation Data

In this study, the observed data regarding hourly integrated land–atmospheric interactions on the TP from 2005 to 2016 were obtained from the Chinese Science Data Center of the TP, which integrates the following six stations: MAWORS, NADORS, BJ, NAMORS, QOMS, and SETORS. Specifically, the following were obtained: hourly meteorological, solar radiation, eddy covariance (EC), and soil moisture, and heat data from the six field sites from 2005 to 2016, including multi-layer gradient observation data composed of wind direction, wind speed, air temperature, relative humidity, precipitation, air pressure, multi-layer soil temperature and moisture data, soil heat flux data, four-component radiation, EC turbulent flux data composed of LE and H, and carbon dioxide flux data [40]. In this study, the data used for analysis were the LE, H, and solar radiation components. The H and LE data were collected using an EC system for observation. The EC systems comprise a sonic anemometer (Campbell, CSAT3) and a fast-response gas analyzer (Li-COR, Li-7500) and were installed at 3.02 m, 2.3 m, 2.75 m, 3.06 m, 3.25 m, and 3.04 m above the ground of BJ, MAWORS, NADORS, NAMORS, QOMS, and SETORS, respectively. The CNR1 and NR01 (Kipp&Zonen) four-component radiation observation systems were used to collect radiation measurements at QOMS, SETORS, NADORS, and MAWORS. The NR01 (Vaisala) four-component radiation observation system was used to measure the Namco station, and the error range was within ±10%. A solar radiation measurement system (CM21, Kipp&Zonen, and PIR, Eppley) was used to measure the surface radiation component; this system can measure shortwave radiation from the surface and longwave radiation from the atmosphere, with error ranges of ±2% and ±5 W/m$^2$, respectively. Local time was used in this study (UTC+8) [40]. The data from 2005 were not included in this study because of discontinuity caused by many missing readings from this year.

#### 2.1.2. ERA-5 Reanalysis Data

The ERA-5 reanalysis data are the fifth generation of global climate atmospheric reanalysis data from the European Center for Medium-Range Weather Forecasts (ECMWF) [41].

ERA-5 combines vast amounts of historical observations into global estimates using advanced modeling and data assimilation systems. The data cover the Earth on a 30 km grid and resolve the atmosphere using 137 levels from the surface up to a height of 80 km. The ERA-5 Land dataset used in this study was a replay of the land component forced by meteorological fields and offers great improvements in precision for land applications [42]. ERA-5 Land dataset coverage was from 1950 to the present time, and was regridded to a spatial resolution of 0.1° × 0.1°. The monthly averaged LE, H, downward and upward radiation, surface albedo, and net radiation from 2006 to 2016 were used in this study.

### 2.1.3. Land-Use Type Data and GVF Data

Land-use type data obtained by MODIS (Terra and Aqua) were also used in this study to assess the underlying surfaces of the TP. The complete MODIS land-use database contains five different land-use datasets: the IGBP dataset [43], the University of Maryland Data Set (UMD) of 14 classes [44], 10 types of MODIS LAI/FPAR algorithm dataset [45], 8 biological datasets [46] and 12 types of plant functional classifications [47,48]. The MODIS data used in this study were obtained from a 21-category IGBP database with a resolution of 5 km.

The GVF was calculated by using MOD13A3 Level 3 monthly 1 km Vegetation Indices data (https://appeears.earthdatacloud.nasa.gov/task/area, accessed on 30 September 2021) and also upscaling to 5 km resolution. The GVF is obtained using the relationship by Gutman and Ignatov (1998) [49]:

$$NDVI = (NDVI - NDVI_{min})/(NDVI_{max} - NDVI_{min}) \tag{1}$$

where $NDVI_{min}$ and $NDVI_{max}$ are bare soil without vegetation (LAI→0) and dense vegetation (LAI→∞), which contain the minimum and maximum NDVI values over the TP, respectively.

### 2.2. Analysis Method

The Formula calculation of *Rn* was as follows:

$$Rn = (Rsd + Rld) - (Rsu + Rlu) \tag{2}$$

In formula (2), the *Rsd* is the downward solar radiation, *Rsu* is the upward radiation, *Rld* is the atmospheric downward longwave radiation, and *Rlu* is the upward longwave radiation (W/m$^2$).

In the error analysis of the ERA-5 data and observation data, as the temporal and spatial resolution of the radiation flux observation data of each station was different from that of the ERA-5, a monthly average processing method was adopted to unify the temporal resolution of all data. Bilinear interpolation was used to interpolate the ERA-5 to the positions of the observation stations. ERA-5 data for the six field sites were obtained using this method. As the ERA-5 monthly mean data are in J/m$^2$ and the cumulative period is 24 h, dividing by the cumulative period expressed in seconds converts the units to W/m$^2$ following the observed data. The formulas for calculating the shortwave and longwave radiation from the land surface upward in the ERA-5 data are as follows:

$$Rsu = Rsd \times fal \tag{3}$$

$$Rlu = Rld - str \tag{4}$$

where *fal* is surface albedo and *str* is surface net thermal radiation.

Three error measures were selected to validate the ERA-5 data: the correlation coefficient (R), bias, and root mean square error (*RMSE*).

R is a statistical indicator that reflects the closeness of the correlation between the variables. The value of R is between −1 and 1. If the coefficient is positive, then the two

variables are positively correlated. If the coefficient is negative, then the correlation is negative. The greater the absolute value, the stronger the correlation.

$$R = \frac{\sum_{i=1}^{n}(x_i - \overline{x})(y_i - \overline{y})}{\sqrt{\sum_{i=1}^{n}(x_i - \overline{x})^2 \sum_{i=1}^{n}(y_i - \overline{y})^2}} \tag{5}$$

The bias describes the difference between the simulated and actual values. In this study, a positive deviation means that the reanalysis data overestimate the observed value, and a negative deviation means that the reanalysis data underestimate the observed value. The calculation formula is as follows:

$$Bias = \frac{\sum_{i=1}^{n}(y_i - x_i)}{n} \tag{6}$$

The *RMSE* is extremely sensitive to the maximum or minimum error response in a set of measurements, so it can better reflect the measurement accuracy. A smaller value indicates a higher accuracy. The calculation formula is as follows:

$$RMSE = \sqrt{\frac{\sum_{i=1}^{n}(y_i - x_i)^2}{n}} \tag{7}$$

where $y_i$ is the predicted value of the reanalysis data, $x_i$ is the observed value, and $n$ is the number of measurements.

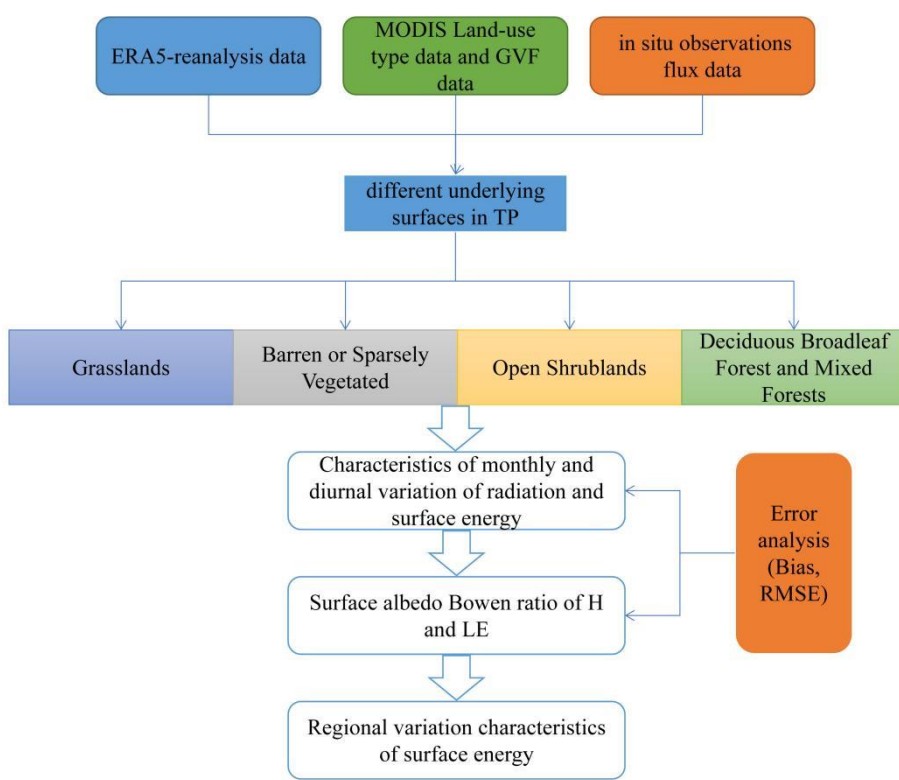

**Figure 1.** Analysis process flow chart.

The data analysis and processing in this study were conducted as follows: first, ERA-5 reanalysis data, MODIS land-use and NDVI data, and flux site observation data were collected, and the data over different underlying surfaces were pre-processed. The underlying surface of the TP was then divided into four main types: Grasslands; Barren or Sparsely Vegetated Lands; Open Shrublands; Deciduous Broadleaf Forest and Mixed Forests. Based on the feedback effect of energy and water on the atmosphere, we analyzed

the monthly variation characteristics of radiation, surface energy flux, the Bowen ratio (β), and surface albedo parameters, and calculated the *RMSE* and bias error. Finally, the distribution characteristics of the ERA-5 data over the TP were obtained, and the applicability of this data was verified. The flow chart of the analysis process is shown in Figure 1, and a schematic diagram of the land-use types, site locations, and elevation on the TP is shown in Figure 2.

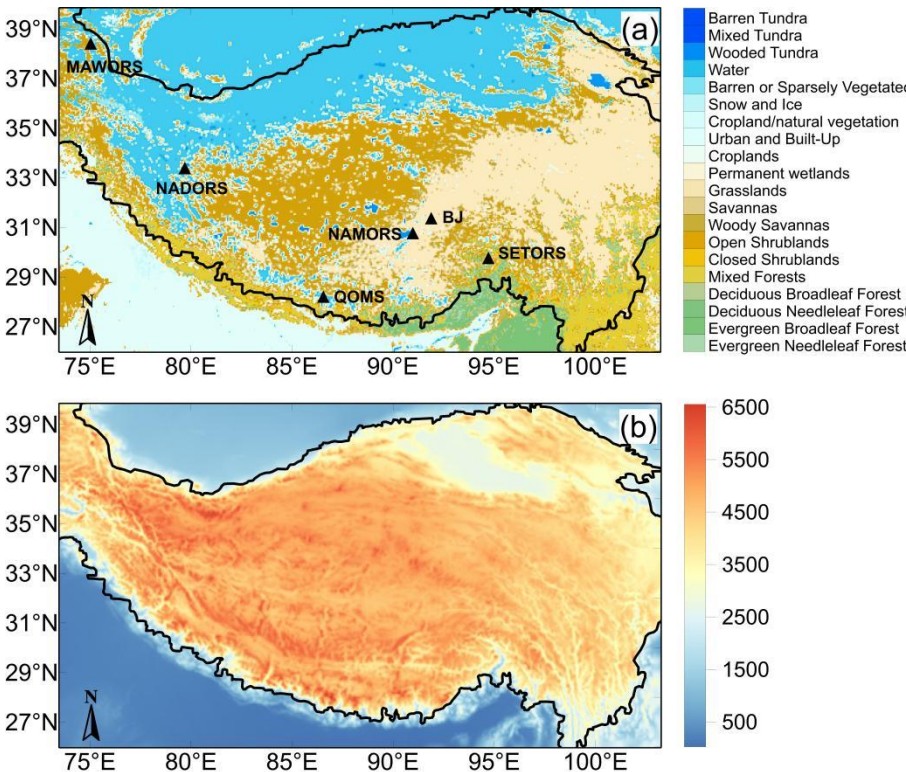

**Figure 2.** (**a**) Distribution of land-use/vegetation types and site locations (symbol) on the TP; (**b**) Elevation diagram on the TP.

## 3. Results and Analysis

### 3.1. Monthly Variation Characteristics of GVF

GVF is an important land surface parameter in land–atmospheric interaction processes and is defined as a part of photosynthetically active green canopy intercepting a midday downward solar grid cell [50]. In the Noah land surface model (LSM), the seasonal variation of GVF also defines the variation of other surface physical characteristics, such as LAI, albedo, roughness length, and surface emissivity [51]. The vegetation distribution is related to precipitation and temperature, and humid and warm areas are conducive to vegetation growth [52]. The annual distribution of GVF gradually decreased from southeast to northwest over the TP. Due to sufficient precipitation and higher temperature in the southeast than in the west TP, vegetation coverage is higher throughout the year. Vegetation is sparse in the northwestern region of the TP. GVF showed an obvious seasonal variation trend, rising in May and gradually decreasing in September (Figure 3e–i). From June to September, the vegetation coverage rate of the TP reached 40−60% (Figure 3f–i).

Of the six sites studied in this paper, MAWORS and NADORS are distributed in the northwest of the TP (Figure 2a), and because of their geographical location, the underlying surface of the two stations is predominantly barren or sparsely vegetated (Table 1), and the GVF is low (Figure 3). The QOMS is located in the south of the TP, and the underlying surface is dominated by barren or sparsely vegetated land (Figure 2a and Table 1). NAMORS and BJ are located in the middle of the TP and the underlying surface is mainly grassland [40]. (Figure 2a and Table 1). As can be seen from Figure 3, their GVF is high from

June to September but low in other months (Figure 3f–i). SETORS is located in the area with the highest annual GVF in the TP (Figures 2a and 3). The underlying surface types are broadleaf forests and mixed forests, which have little influence on seasonal changes.

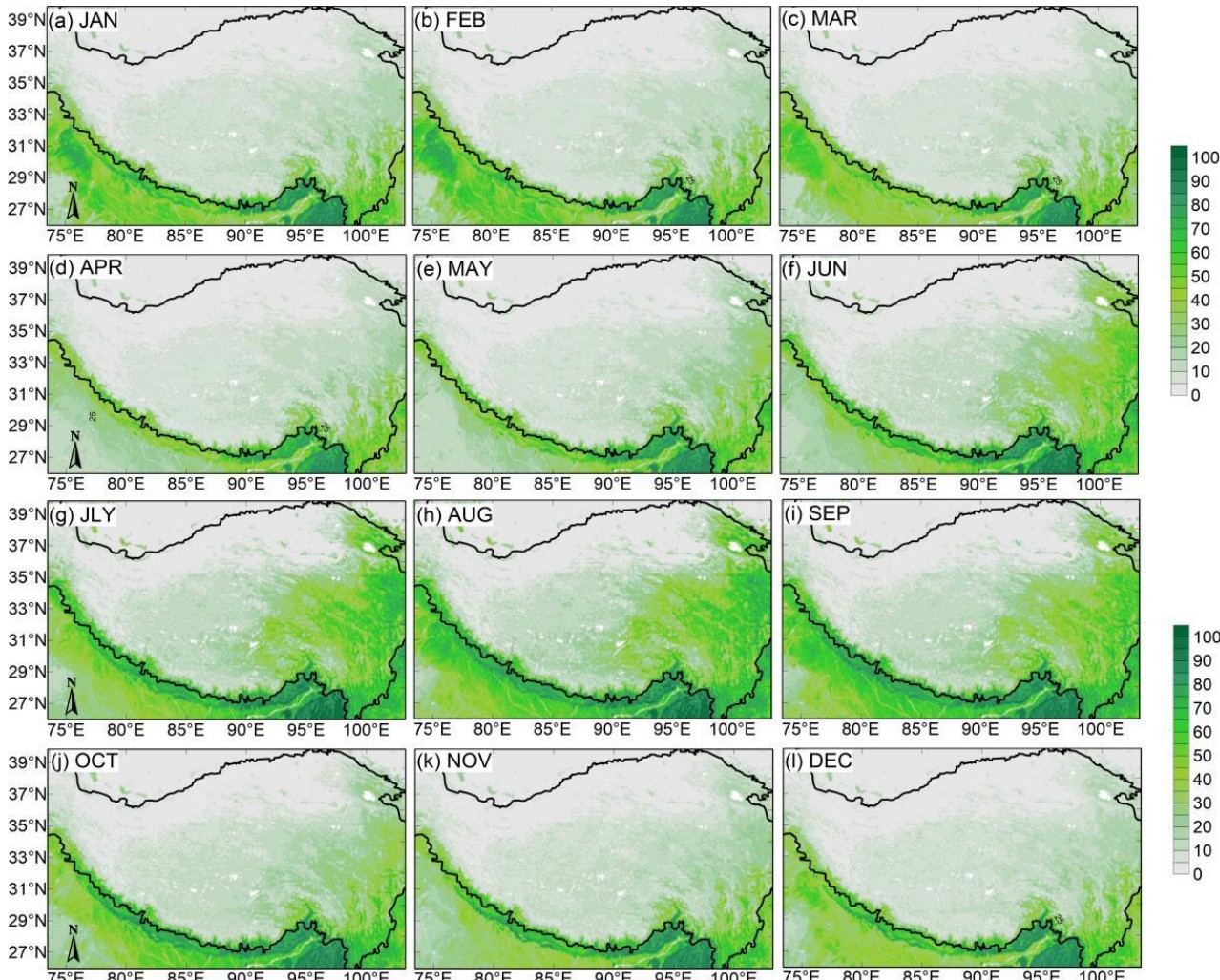

**Figure 3.** Spatial distribution characteristics of green vegetation fractions in different months on the Tibetan Plateau (Unit: %).

*3.2. Variation Characteristics of Surface Energy*

3.2.1. Seasonal Variation Characteristics of Surface Energy

The monthly energy variation characteristics of the six stations were different, but also had some similarities. In this study, the four seasons were divided as follows: spring from March to May, summer from June to August, autumn from September to November, and winter from December to the following January. The highest values of H were observed in the spring, decreased in summer, increased to varying degrees in the autumn, and decreased again in winter. After each station's H achieved its maximum value in spring, the time at which it started decreasing varied, with the SETORS station being the earliest. The LE showed a unimodal change. Before the outbreak of the southeast monsoon on the TP, the LE value was minimal. Precipitation rose in summer, soil moisture increased, latent heat exchange was intense, and LE increased rapidly, with the maximum value exceeding $100 \, \mathrm{W/m^2}$ (Figure 4a). In autumn, it gradually decreased and reached a minimum value during winter. The difference between H and LE at NADORS and QOMS stations in summer was smaller than for the other four stations (Figure 4c,e). The four-component radiation data showed a single-peak variation, and the solar shortwave downward ra-

diation began to decrease in summer. The *Rsu* values at BJ, QOMS, and NAMORS in spring and winter increased with different amplitudes (Figure 4a,d,e). Longwave radiation increased in spring and summer and decreased in autumn and winter. *Rn* increased in spring, reached a maximum in summer, and decreased in autumn and winter. Possible errors were observed in the longwave radiation values at SETORS, meaning these data were not taken into consideration in analysis.

**Table 1.** Description of geographic features of six sites.

| Site | Latitude, Longitude | Elevation (m) | Land Cover | Initial Observation Time of the Instrument (Radiations/EC) |
|---|---|---|---|---|
| BJ | 31.37° N, 91.90° E | 4509 | Grasslands | 2006 |
| MAWORS | 38.41° N, 75.04° E | 3668 | Barren or Sparsely Vegetated and Open Shrublands | 2010 |
| NADORS | 33.39° N, 79.70° E | 4270 | Barren or Sparsely Vegetated | 2009/2005 |
| NAMORS | 30.77° N, 90.99° E | 4730 | Grasslands | 2005 |
| QOMS | 28.21° N, 86.56° E | 4298 | Barren or Sparsely Vegetated | 2005/2007 |
| SETORS | 29.77° N, 94.73° E | 3327 | Deciduous Broadleaf Forest and Mixed Forests | 2007 |

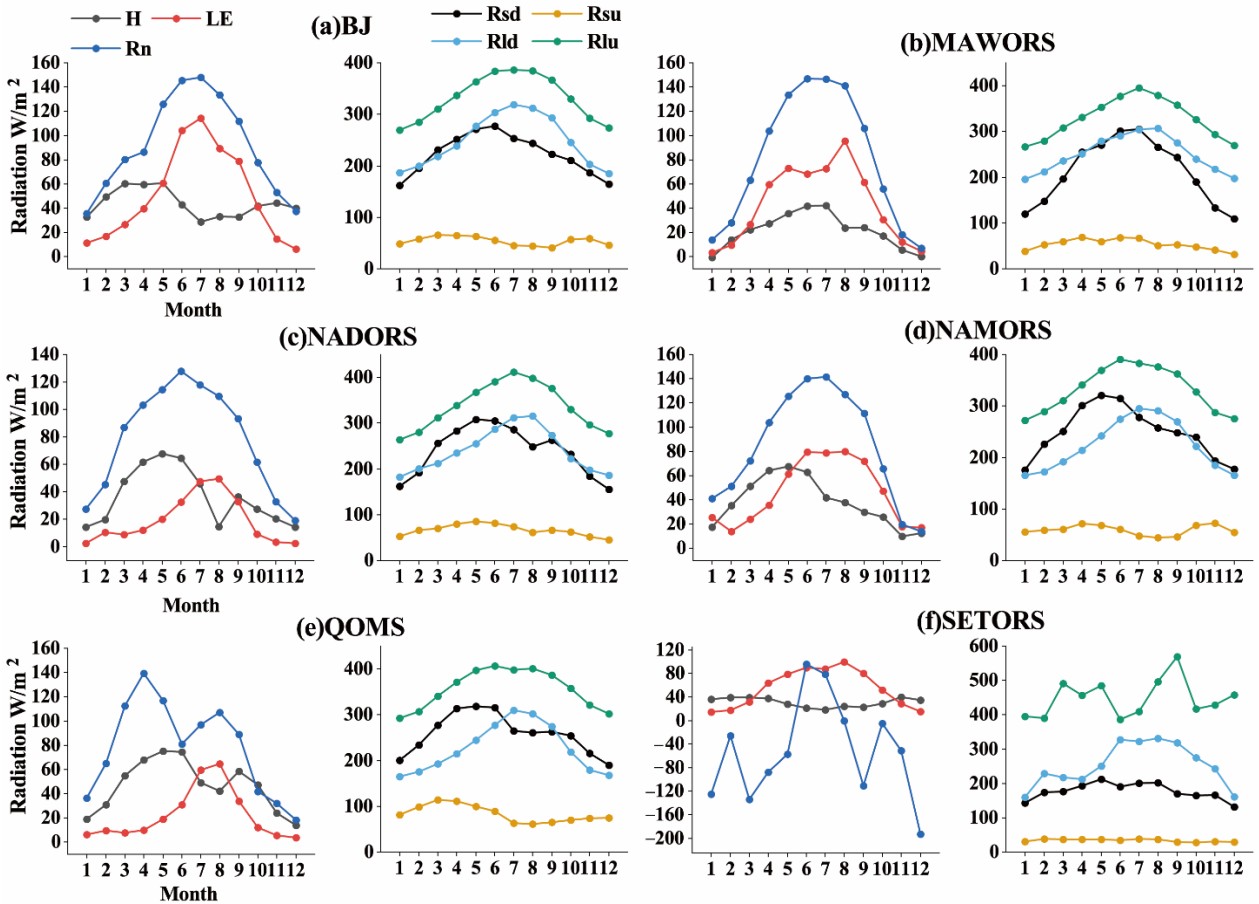

**Figure 4.** Annual average monthly variation of surface energy at six sites.

### 3.2.2. Diurnal Variation Characteristics of Surface Energy

To illustrate the diurnal variation of surface energy fluxes in different regions of the TP, Figures 5 and 6 show the diurnal variation of H, LE, *Rn*, and four-component radiation at the six stations in summer and winter. As shown in Figure 5, all three variables reached

their peak values at approximately 14:00. Sunshine is highest when the solar altitude is high, and the surface obtains more energy. In summer, *Rn* increased gradually from 7:00 to 9:00, peaked at approximately 14:00, and decreased to its lowest value in a day at 23:00. The peak value of *Rn* in summer was more significant than that in winter, with a difference of approximately 250 W/m$^2$. The variation in H and LE was the same as that of *Rn*, and the maximum LE could be more than 200 W/m$^2$. The diurnal variations of H and LE increased at sunrise and decreased at sunset. The LE was generally greater than H in the summer. However, the opposite was true for NADORS and QOMS (Figure 5c,e) because the underlying surface of the two stations comprises barren or sparsely vegetated land, meaning the latent heat exchange is not intense, resulting in an LE that is lower than H. The maximum difference in H and LE between BJ and SETORS could reach more than 100 W/m$^2$ (Figure 5a,f) because the underlying surface of both stations is covered by dense vegetation, with high precipitation and high soil moisture. The diurnal variation trend of each variable at each station in winter was the same as that in summer, but the peak values at all three stations were lower than those in summer. H was higher than LE in winter because the plateau area was in the non-growing period, and vegetation was reduced.

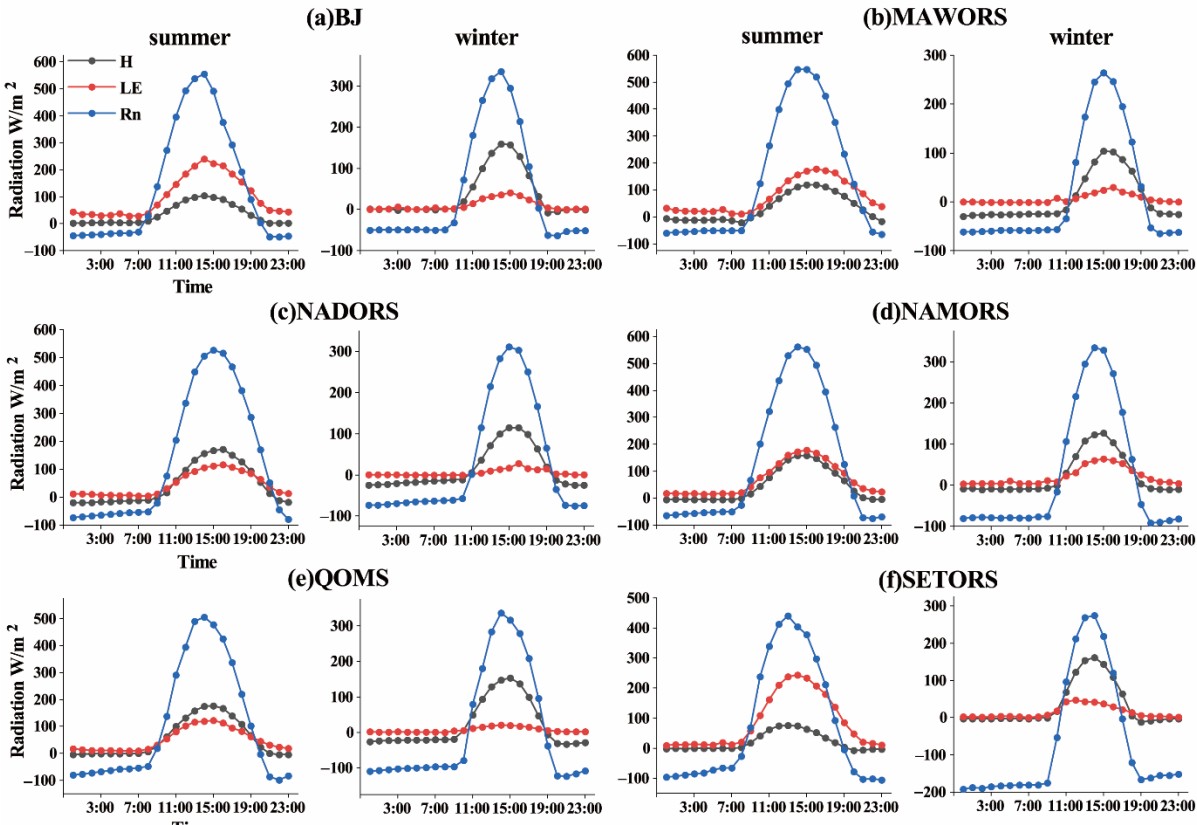

**Figure 5.** Diurnal variation of monthly mean H, LE, and *Rn* in summer and winter.

Figure 6 shows the diurnal variation of the four-component radiation at the six stations in summer and winter. It can be observed from Figure 6 that the variation trends of the four-component radiation in summer and winter were the same. While *Rld* did not vary significantly with time during summer, the other three variables all increased at sunrise and reached a peak at approximately 14:00, then gradually decreased and reached their lowest levels at 23:00. The surface heat was mainly obtained from *Rsd*, reaching a maximum of 900 W/m$^2$ or more at noon. There was a significant difference between *Rsd* and *Rsu* in the summer, with a maximum of approximately 700 W/m$^2$ (Figure 6e). However, the difference decreased in winter, with a maximum of approximately 600 W/m$^2$, due to the reduced solar radiation in winter. The difference between *Rlu* and *Rld* was smaller than

that for shortwave radiation, about 100 W/m$^2$, while the difference was approximately 200 W/m$^2$ at SETORS (Figure 6f).

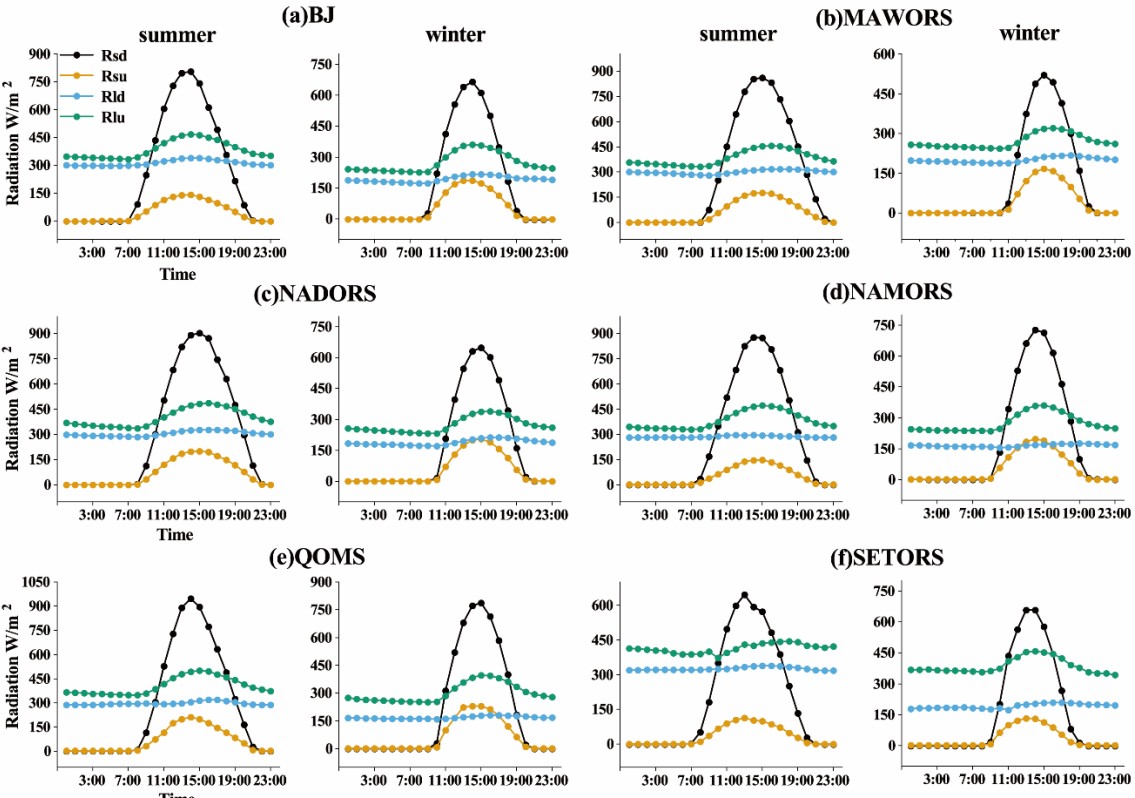

**Figure 6.** Diurnal variation of monthly mean four-component radiation in summer and winter at six sites.

### 3.3. Surface Energy Budget and Distribution

#### 3.3.1. Land Surface Albedo

Land surface albedo is an essential factor affecting the surface energy budget and distribution, and is mainly determined by two factors: the underlying surface conditions and solar altitude. The albedo was calculated using observational data from 8:00 to 20:00 LT. As shown in Figure 7, the variation of the surface albedo presents as a "U" shaped curve, and was higher in the morning and evening and lower at noon. The solar altitude angle was higher at noon and the surface reflected a minor level of *Rsd*. When the solar altitude angle is low, longwave radiation makes up a major part of the solar radiation that reaches the earth surface. The land surface is highly reflective of longwave radiation. The lowest value varied between 0.2 and 0.4 at each station, and the albedo change at each station differed with the season. Whereas, at NADORS and SETORS, the seasonal variation was relatively insignificant (Figure 7c,f), the surface albedo at the other stations gradually decreased from January to May, reaching the lowest values in July or August, and then gradually increased. The main reason for this is that the albedo rises in winter due to heavy snow cover but falls in spring and summer when the snow melts and vegetation grows, resulting in a decreased albedo. For MAWORS and QOMS (Figure 7b), the underlying surface is barren or sparsely vegetated and the surface albedo should be high. However, there was still sparse vegetation growth in summer, which may be the reason for the decrease in the surface albedo of these two stations in summer. SETORS maintains a low surface albedo throughout the year because of the lush vegetation coverage and little influence of seasonal variation on the station (Figure 7f).

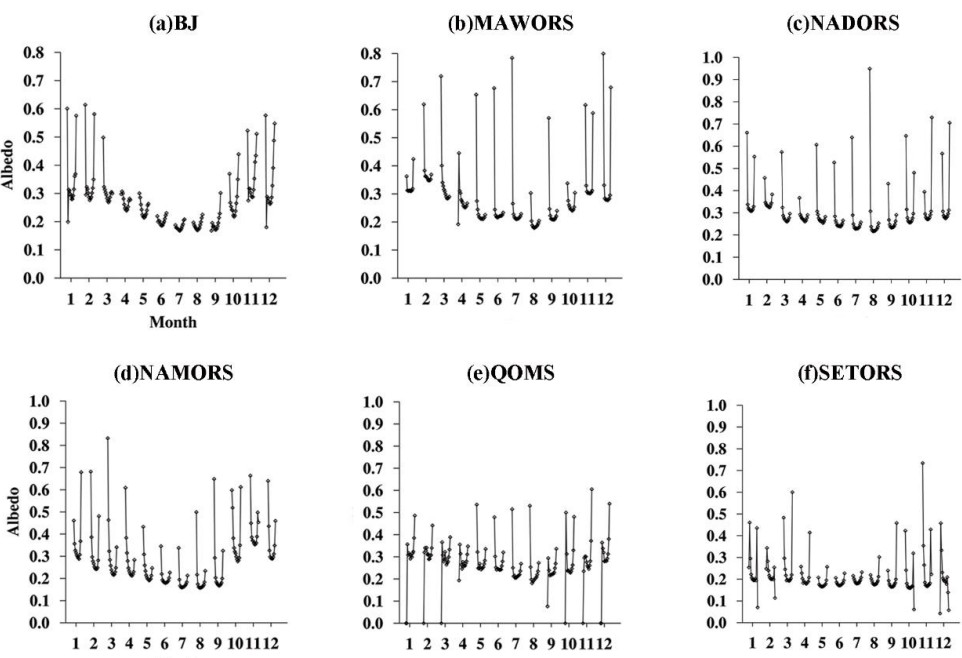

**Figure 7.** Diurnal variation of monthly mean surface albedo at six sites.

### 3.3.2. Surface Energy Distribution

To further illustrate the effects of the different routes of energy transfer on the different underlying surfaces, Figures 8 and 9 show the distribution of the surface energy. As can be seen from Figure 8, latent heat played a leading role in energy exchange in BJ, MAWORS, NAMORS, and SETORS (Figure 8a,b,d,f), while sensible heat was the dominant source of surface energy exchange in NADORS and QOMS (Figure 8c,e). Since the underlying surface of BJ and SETORS has dense vegetation, the transpiration of plants was more notable than the soil heat source effect, and latent heat was the main source of energy transfer [53]. For some years, the primary route of energy transfer at NAMORS was sensible heat, and plant transpiration was less than the soil heat source effect, resulting in a weaker latent heat exchange versus sensible heat exchange.

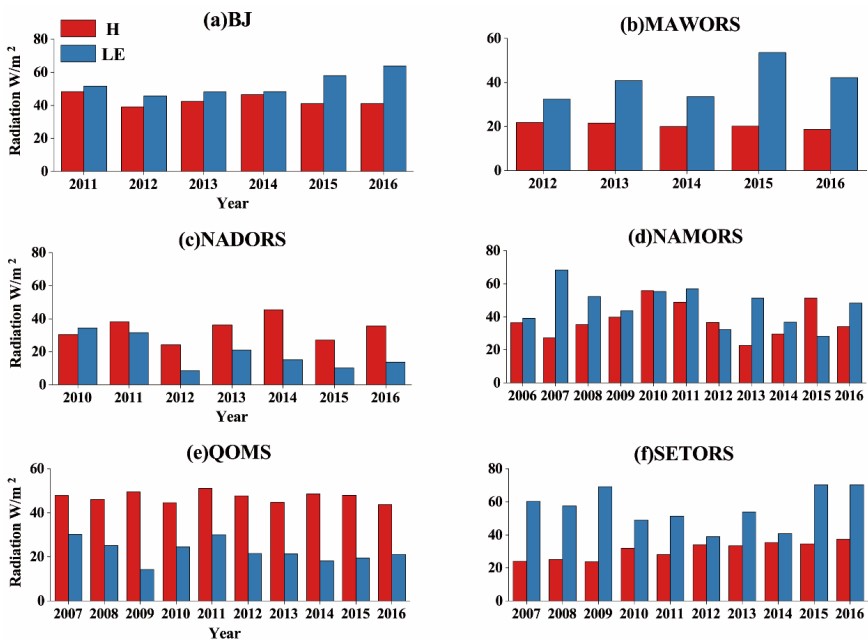

**Figure 8.** Comparison of annual mean values of H and LE at six sites.

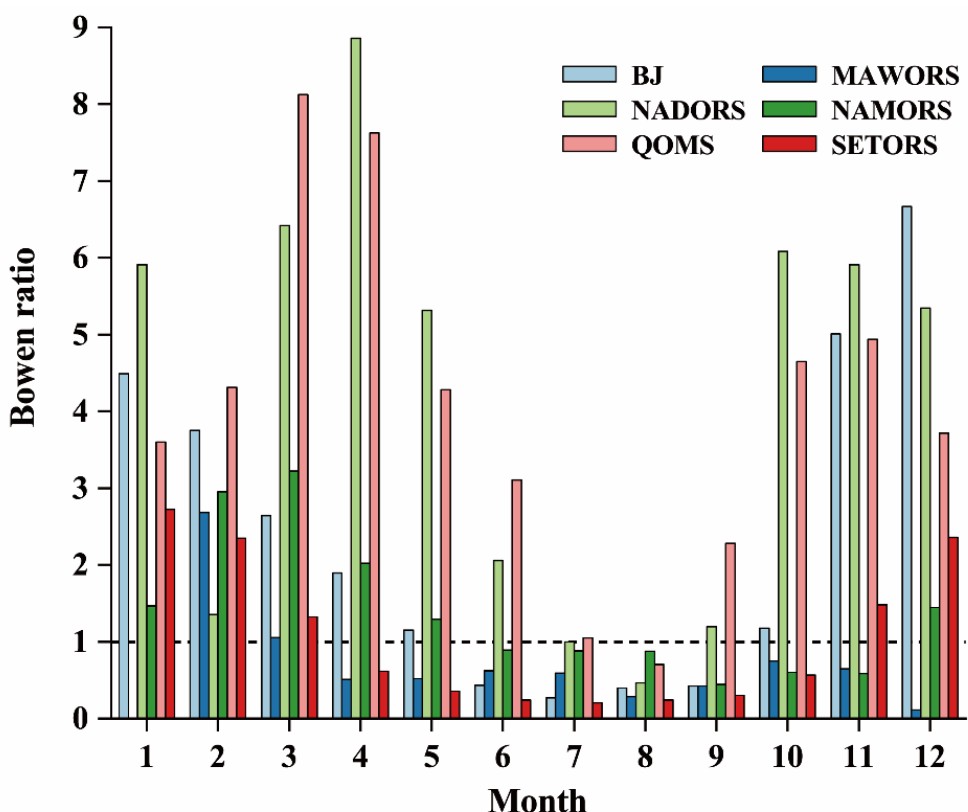

**Figure 9.** Monthly variation of Bowen ratio (H/LE) at six sites (dashed line indicates that H = LE).

The Bowen ratio (β) is defined as the ratio of H to LE at the surface. A higher value of β indicates greater sensible heat exchange; otherwise, the latent heat exchange is higher. As shown in Figure 9, sensible heat exchange was the main form of energy transfer in winter at all stations except MAWORS. The main reason for this is that plants are in the non-growing phase in winter, resulting in a decline in vegetation and soil hydrothermal conditions [54]. The seasonal variation trends in BJ and SETORS were identical, decreasing in spring and increasing in autumn. Latent heat was the main energy distribution process in BJ in summer, whereas sensible heat was the main process during the other seasons. The variation range at SETORS was smaller than that of BJ, and the energy distribution was mainly latent heat, except in winter. The heat exchange between the land surface and the atmosphere at MAWORS was dominated by latent heat exchange throughout the year, which is consistent with the comparison of the annual mean values of LE and H. β was less than 1 at NADORS during July and August only, and greater than 1 for all other months. The maximum value reached was 8.8, indicating that the energy transfer at NADORS occurred mainly via sensible heat exchange, which accounted for a large proportion. The β levels were relatively seasonally balanced at NAMORS. The monthly variation of β at QOMS was the same as that at NADORS because the underlying surface is mainly desert with sparse vegetation and weak latent heat exchange.

### 3.4. Error Analysis of ERA-5 Land Data and Observation Data

As the ERA-5 data are reanalysis data, they can be affected by many factors that may cause them to deviate from actual observation values. Error analysis of the ERA-5 data and the observation data from the six stations was carried out to determine whether the ERA-5 data had a high degree of accuracy and could be used to study the surface energy changes of the entire TP region.

Table 2 lists the R between the observed data from the six stations and the ERA-5 data. We can see that, of all the variables, *Rld* has the highest correlations, almost all above 0.9, followed by *Rsd*. H exhibits the lowest correlation, mostly under 0.5, followed by *Rsu*.

Of all the stations, the correlation between the ERA-5 data and the observed values is highest at MAWORS, and lowest at SETORS, and H is actually negatively correlated. The low correlation may be attributed to underlying surface conditions. Actual underlying surface conditions are more complex than those simulated by the ERA-5 data, leading to uncertainty in the predicted values.

**Table 2.** R between ERA-5 data and observed values (* indicates a failure to pass the significance test).

| Variable \ Site | BJ | MAWORS | NADORS | NAMORS | QOMS | SETORS |
|---|---|---|---|---|---|---|
| *Rsd* | 0.94 | 0.95 | 0.96 | 0.70 | 0.79 | 0.57 |
| *Rsu* | 0.39 | 0.52 | 0.44 | 0.31 | 0.60 | 0.26 |
| *Rld* | 0.94 | 0.98 | 0.98 | 0.97 | 0.98 | 0.41 |
| *Rlu* | 0.95 | 0.96 | 0.94 | 0.87 | 0.79 | 0.11 * |
| H | 0.49 | 0.77 | 0.56 | 0.37 | 0.62 | −0.50 |
| LE | 0.90 | 0.76 | 0.81 | 0.41 | 0.77 | 0.84 |
| *Rn* | 0.85 | 0.91 | 0.90 | 0.78 | 0.30 | 0.17 * |

Surface heat flux is mainly limited by soil temperature and moisture (vegetation cover, atmospheric conditions, and soil physical characteristics). Soil temperature and moisture are greatly affected by precipitation, especially in arid regions [55–57]. Figure 10 shows the monthly variation in the bias between the ERA-5 reanalysis data and observations. It can be seen from the picture that the longwave radiation values of the ERA-5 data underestimated the observed values (except for SETORS) (Figure 10c,d). The bias variation ranges of H and *Rld* are relatively minimal, and the variation range is within 60 W/m$^2$ (Figure 10c,e). All stations show the same variation in the shortwave radiation bias during spring and summer, which becomes larger in spring and smaller in summer, indicating that the predicted value gradually approaches the observed value in summer, and then reaches a minimum in autumn and winter (Figure 10a,b). The bias of *Rld* did not change significantly with time (except at SETORS). The bias values are approximately 50 W/m$^2$ lower than the observed values at MAWORS and NADORS, while they are approximately 30 W/m$^2$ lower at BJ, QOMS, and NAMORS, among which NAMORS shows the smallest bias (Figure 10c). The bias of *Rlu* changes significantly in spring and summer, but little in autumn and winter (Figure 10d). The bias of H was large in the first five months, reached a maximum bias in March at most stations, and then gradually decreased (Figure 10e). Precipitation uncertainty leads to a bias between the surface and soil moisture, leading to a greater uncertainty regarding the LE levels in the ERA-5 data. The predicted LE is closest to the observed value in summer because there is more precipitation and high soil moisture on the TP at this time, and the uncertainty of the ERA-5 data is decreased (Figure 10f). *Rn* showed a high underestimation to different degrees from January to June. There was little change at other stations from July to December, except for QOMS, which had a relatively high estimate in November (Figure 10g). As mentioned above, there are some errors in the longwave radiation data from SETORS; these resulted in an abnormal bias fluctuation, which will not be discussed here.

As *RMSE* can better reflect the accuracy of data, this study also used *RMSE* as an index to evaluate the accuracy of the ERA-5 data. Figure 11 shows the monthly variation in the *RMSE* of the ERA-5 data and the observed data. Generally speaking, the accuracy is highest for BJ and lowest for MAWORS. As the MAWORS is located in a barren or sparsely vegetated area, the precipitation is low, and the soil temperature and soil moisture are greatly affected by precipitation, which adds more uncertainty to the ERA-5 data. The accuracy is higher for summer and lower for spring at most of the stations. The *RMSE* of *Rld* does not change significantly with time (Figure 11c), but the RMSE of LE and H do (Figure 11e,f). The *RMSE* changes in shortwave radiation, *Rlu*, and *Rn* are the same, and their accuracy is higher for summer (Figure 11a,b,d,g).

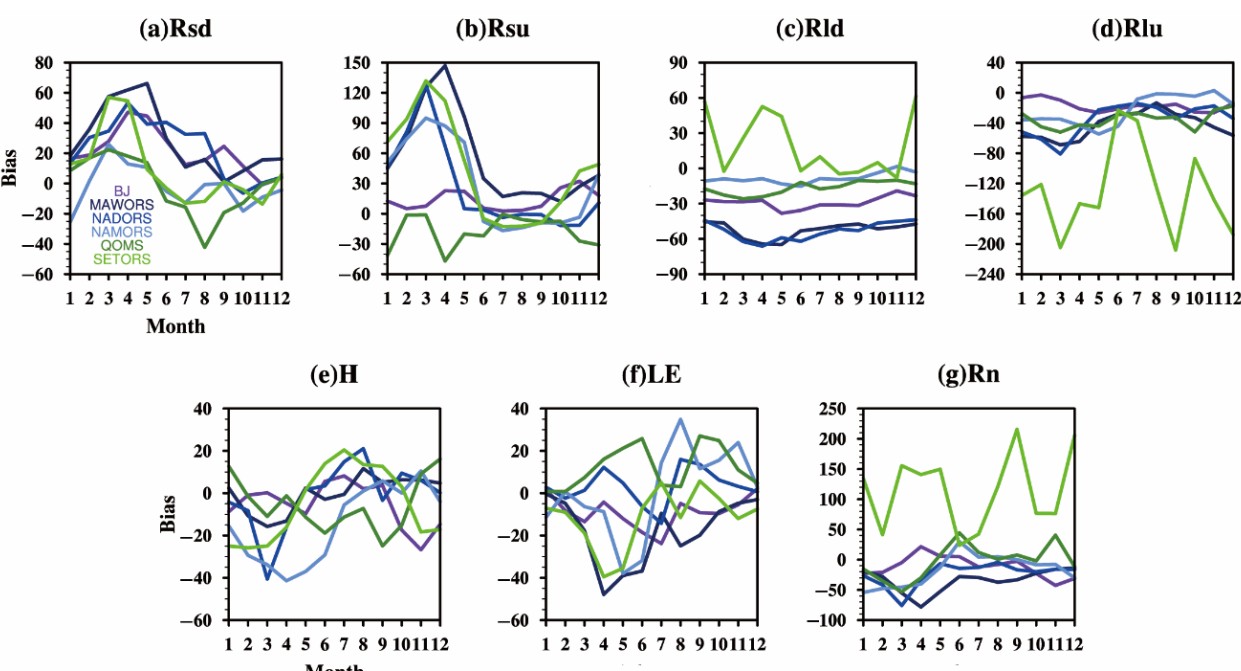

**Figure 10.** Monthly change of Bias between ERA-5 data and observed values (Unit: W/m$^2$).

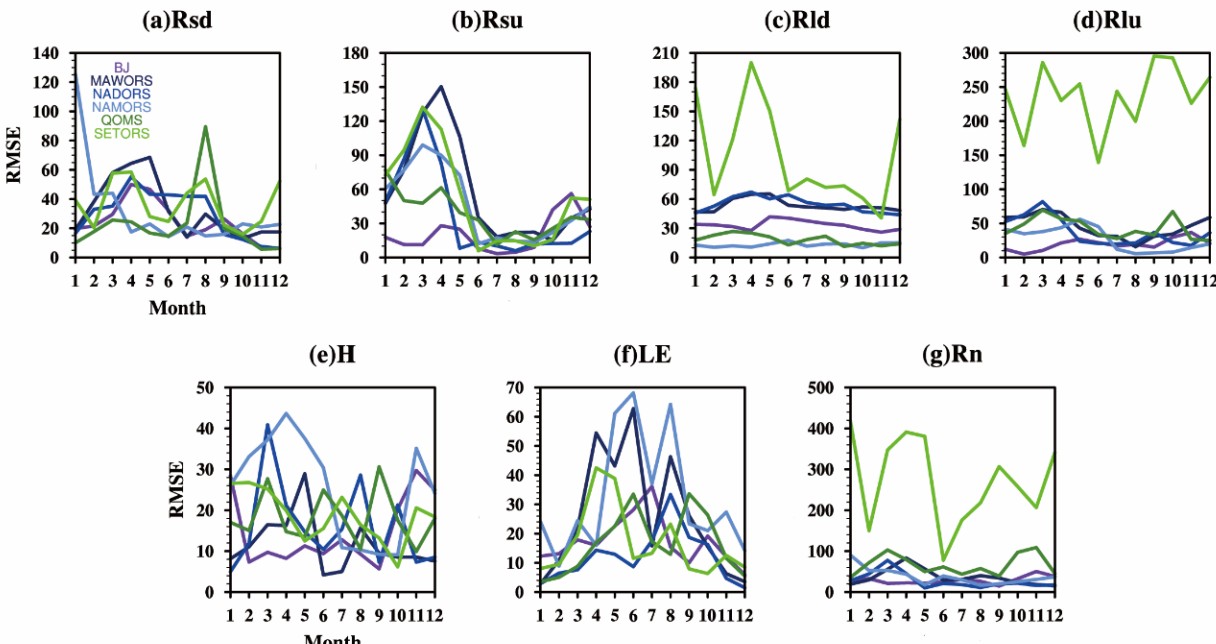

**Figure 11.** Monthly change of *RMSE* between ERA-5 data and observed values (Unit: W/m$^2$).

*3.5. Energy Variation Characteristics of Surface Area*

The changes in surface energy over time at the six stations were analyzed in the previous section. This section considers the changes in regional energy across the TP. Figure 12 shows the seasonal variation in the surface energy of the TP. It can be seen that the LE had an obvious seasonal variation, and the LE in the north and west of the plateau could reach more than 60 W/m$^2$ in spring and autumn. In most other locations, it was 20–40 W/m$^2$ (Figure 12a,c). However, in summer, the area with the highest values of LE was primarily located in the east of the plateau, where the maximum could exceed 80 W/m$^2$ (Figure 12b). This is due to the onset of the summer monsoon, resulting in higher precipitation and lush vegetation in the east of the plateau, and intense latent heat exchange

between the surface and the atmosphere, which does not occur in winter. In spring and summer, the area with the highest values of H was mainly concentrated in the west of the plateau, with an average of 40–60 W/m$^2$ (Figure 12e,f), indicating that the turbulent movement in the west of the plateau was relatively strong at this time. *Rn* reached its peak in summer (Figure 12j), and *Rn* in the north of the plateau was always higher than that in the south, except in summer (Figure 12i,k,l). Generally, the LE was higher in the north and east of the plateau, whereas the maximum H was mainly in the west of the plateau. The energy value of H had a smaller variation range over time compared with that of LE. The *Rn* value in the north of the plateau was higher than that in the south (except in summer).

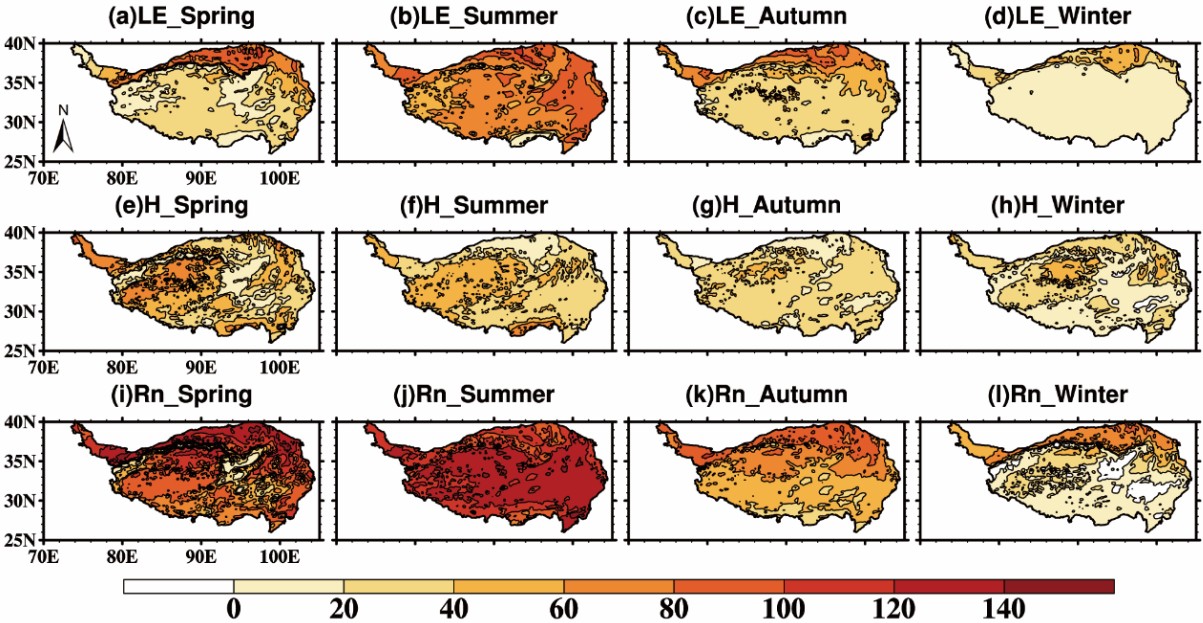

**Figure 12.** Annual average seasonal variation of ERA-5 data (LE, H, and *Rn*).

Figure 13 shows the seasonal variation in four-component radiation over the TP. As shown in the figure, *Rsd* reached its maximum value in spring, and the high-value area was mainly located in the western and central parts of the TP. At this time, the average *Rsd* on the TP was approximately 300 W/m$^2$ (Figure 13a). The high-value area was reduced during the summer, during which time the *Rsd* of the plateau was approximately 250 W/m$^2$ on average (Figure 13b). The radiation value gradually decreased in autumn and winter (Figure 13c,d). However, the decrease in the west of the plateau was smaller than that in the eastern part of the plateau. The *Rsu* value was relatively low throughout the year, between 50 and 100 W/m$^2$, while that in the north of the plateau was lower than 50 W/m$^2$ (Figure 13e–h). When combined with Figure 2a, it can be seen that the northern part of the underlying plateau surface is complex, and *Rsd* is relatively low, which may be the reason for the low value of *Rsu* in the north of the plateau. The value of longwave radiation was generally larger than that of shortwave radiation on the plateau. For *Rld*, the seasonal variation was obvious. It increased in spring and reached its maximum in summer, and the radiation value varied between 250 and 300 W/m$^2$ (Figure 13i,j), decreasing gradually in autumn and winter (Figure 13k,l). The annual value of *Rlu* was greater than that of the first three variables, and the seasonal variation was similar to that of *Rld*. It reached its peak in summer, with levels of more than 350 W/m$^2$ in most of the plateau and more than 450 W/m$^2$ in the west of the plateau (Figure 13n). The *Rlu* value in the south of the plateau was always lower than that in the north (Figure 13m–p).

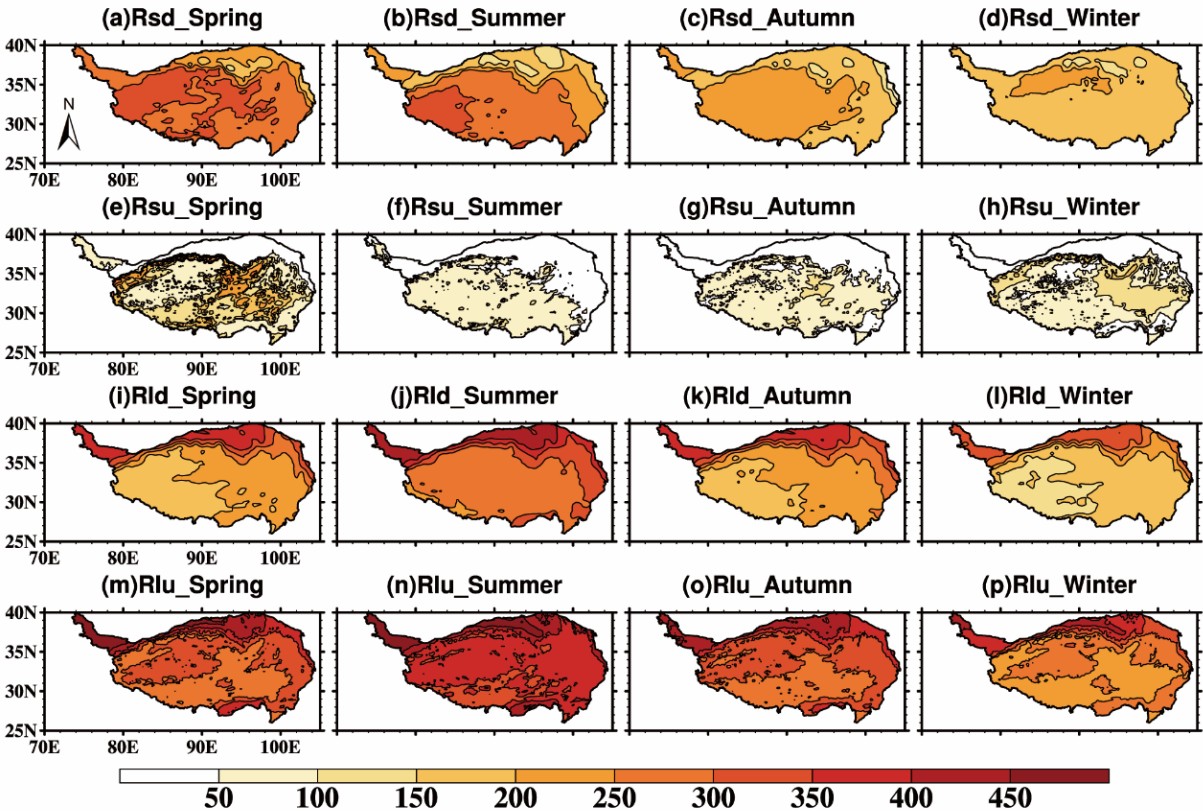

**Figure 13.** Annual average seasonal variation of ERA-5 four-component radiation (*Rsd*, *Rsu*, *Rld*, and *Rlu*).

## 4. Discussion

In this paper, MODIS land-use data and NDVI data were used to acquire the underlying surface vegetation types and analyze the distribution of the seasonal variation of GVF over the TP. The geographical location and underlying surface conditions had a great impact on the exchange of surface energy flux. In general, during the vegetation growth period on the TP, the three stations with a higher GVF (BJ, NAMORS, and SETORS), recorded a lower surface albedo, resulting in a decrease in *Rsu* and an increase in *Rn*. The radiation energy was absorbed by the large number of plants and by the soil. Moreover, evaporation from the land surface and vegetation increased, resulting in intensive latent heat exchange. The LE increased rapidly in summer, and played a leading role in surface energy transfer. However, in the low GVF areas (NADORS and QOMS), the surface albedo was always high, causing the surface energy exchange to be dominated by sensible heat. We also found that the relationship between energy distribution and the underlying surface in the MAWORS site area was different from the above mentioned. The MAWORS station is located in the west of the TP, and the underlying vegetation is sparse, but the value of LE was always higher than H throughout the year. The reasons for this phenomenon need further study. After comparative analysis with the observational data, we found that the ERA-5 data have good applicability in the TP. The discrepancy between the ERA-5 radiation data and underlying surface energy flux data was higher in spring and lower in summer over the TP. We preliminarily analyzed the surface radiation and energy variation characteristics of six flux sites in different regions of the TP, and considered the impact of underlying vegetation coverage and land-use types on the energy distribution. However, the contribution of different regions' energy transfer ratios needs to be further examined. The use of ERA-5 reanalysis data to analyze the differences in energy distribution in different regions of the TP also requires further research.

## 5. Conclusions

Based on the observational data from six stations (BJ, MAWORS, NADORS, NAMORS, QOMS, and SETORS) on the TP, the surface energy variation and energy distribution were studied. The ERA-5 Land data were used to study the regional energy changes of the TP after error analysis, and the following results were obtained:

(1) The annual distribution of GVF gradually decreased from southeast to northwest over the TP. Owing to the influence of precipitation and temperature, vegetation coverage in the southeastern TP is relatively high throughout the year. From June to September, the vegetation coverage rate of the TP reached 40−60%.

(2) Monthly variations in surface energy characteristics included the following. H increased in spring and autumn and decreased in summer and winter. After H reached its maximum value in spring, the decrease began at different times at each station, and was earliest at the SETORS station. The LE increased rapidly in summer, with a maximum value of more than 100 $W/m^2$, and gradually decreased in autumn and winter. In summer, the difference between H and LE at the NADORS and QOMS stations was lower than that at the other four stations. The four-component surface radiation increased during spring and summer, and decreased in autumn and winter.

(3) The diurnal variation in the surface energy obeyed the following trends. Except for *Rld*, which changed insignificantly over time, these variables began to increase at sunrise, reached their maximum values at noon, and decreased at sunset. LE was generally greater than H in summer, but the opposite was true for NADORS and QOMS. In winter, H was generally greater than LE. Longwave radiation differs from shortwave radiation in that it is more susceptible to solar radiation.

(4) The surface albedo changed in a "U" shape curve, and was high in the morning and evening, and low at noon. Except for NADORS and SETORS, where the surface albedo changed insignificantly with the seasons, all stations showed a gradual decrease in spring, reached their lowest values in summer, and gradually increased in autumn and winter. The interannual variation in H and LE shows that latent heat exchange is the main form of energy transfer in BJ, MAWORS, NAMORS, and SETORS. In contrast, sensible heat played a leading role in surface energy transfer at NADORS and QOMS. The Bowen ratio was generally low in summer, and some sites had a maximum in spring.

(5) The *Rld* value of ERA-5 at each station had the highest correlation with the observed value. The longwave radiation value of ERA-5 was lower than the observed value, and the bias of the shortwave radiation increased in spring and decreased in summer. Among the six stations, the highest precision was observed for BJ.

(6) The LE increased in spring and summer and decreased in autumn and winter, with the highest levels mainly concentrated in the north and east of the plateau (during summer). The high-value area of H was mainly in the west of the plateau. When *Rn* varied with the season, the radiation value in the north of the plateau was always higher than that in the south of the plateau (except in summer). The four components varied significantly with the seasons. *Rld* in the east of the plateau was higher than that in the west, and *Rsd* in the east of the plateau was lower than that in the west. The maximum *Rlu* values were in the northwest and northeast of the plateau.

**Author Contributions:** Conceptualization, X.W. and J.M.; methodology, X.W.; software, X.W.; validation, J.M., M.L. and S.L.; formal analysis, J.M.; investigation, J.M.; resources, X.W., X.Z., X.Y. and M.C.; data curation, X.W.; writing—original draft preparation, J.M.; writing—review and editing, X.W.; visualization, J.M.; supervision, X.W.; project administration, X.W.; funding acquisition, X.W. All authors have read and agreed to the published version of the manuscript.

**Funding:** This research was funded by the Second Tibetan Plateau Scientific Expedition and Research Program (STEP) (grant number 2019QZKK010304), the National Natural Science Foundation of China (grant number 41975096), and the Innovation Team Fund of Southwest Regional Meteorological Center, China Meteorological Administration (no grant number).

**Data Availability Statement:** A long-term (2005-2016) dataset of hourly integrated land–atmosphere interaction observations on the Tibetan Plateau (https://doi.org/10.5194/essd-12-2937-2020, accessed 20 July 2021). ERA5-Land data (10.24381/cds.68d2bb30, accessed on 15 September 2021). MOD13A3 Level 3 monthly 1 km vegetation indices data (https://appeears.earthdatacloud.nasa.gov/task/area, accessed on 30 September 2021).

**Acknowledgments:** The authors thank the Institute of Tibetan Plateau Research, CAS, for the long-term (2005–2016) radiation observation data, and the ECMWF for the ERA5-Land reanalysis data that provided support for this study.

**Conflicts of Interest:** The authors declare no conflict of interest.

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
