# Peer review of "Analysis of Surface Energy Changes over Different Underlying Surfaces Based on MODIS Land-Use Data and Green Vegetation Fraction over the Tibetan Plateau"

_remotesensing, doi:10.3390/rs14122751_

Round 1
Reviewer 1 Report
The paper is well structured and interesting, need to consider following comments before publishing.
- Try to provide methodology flow chart.
- North arrow missing in all maps.
- make it Table 1 more clear.
- Quality of all graphs should be improved.
- Spell and grammar check is needed.
Author Response
Response to Reviewer 1 Comments
Point 1: Try to provide methodology flow chart.
Response 1: Thank you for your suggestion. A flowchart of the analysis has been included in the manuscript. Please refer to page 5, lines 219–231 of the manuscript, which are as follows:
The data analysis and processing in this study were as follows: first, ERA-5 reanalysis data, MODIS land use and NDVI data, and flux site observation data were collected, and the data over different underlying surfaces were pre-processed. The underlying surface of the TP was then divided into four main types: Grasslands, Barren or Sparsely Vegetated Lands, Open Shrublands and Deciduous Broadleaf Forest, and Mixed Forests. Based on the feedback effect of energy and water on the atmosphere, we analyzed the monthly variation characteristics of radiation, surface energy flux, Bowen ratio (β), and surface albedo parameters, and calculated the RMSE and bias error. Finally, the distribution characteristics of ERA-5 data over the TP were obtained, and the applicability of this data was verified. The flow chart of the analysis process is shown in Figure 1.
Figure 1. Analysis process flow chart.
- North arrow missing in all maps.
Response 2: Thank you for raising this point. The north arrow has been added to Figures 2, 3, 12, and 13. Please see these figures on lines 233 (page 6), 263 (page 7), 484 (page 17), and 486 (page 17) respectively.
- make it Table 1 more clear.
Response 3: Thank you for your comment. As per your suggestion, we have reformatted Table 1 to improve the clarity. Please refer to page 8, line 265.
- Quality of all graphs should be improved.
Response 4: Thank you for your suggestion. We have duly revised figures 4, 5, 6, 10, 11, 12, and 13. Please refer to these figures in the manuscript on pages 9, 10, 11, 15, and 17 respectively.
- Spell and grammar check is needed.
Response 5: Thank you for raising this point. We have carefully checked for spelling and grammar throughout the manuscript.

Reviewer 2 Report
See applied file

Author Response
Response to Reviewer 2 Comments
Point 1: Though the investigation is connected with defifinite locality its methodology and results have general character and represent doubtless interest for specialists in atmosphericphysics, agroculture and climatology. The article is written suffiffifficiently clear and can be published. In (6) parentheses must be added.
Response 1: Thank you very much for your suggestion. the parentheses have been added in formula (6). Please see line 210 in manuscript.
(6)

Reviewer 3 Report
The manuscript has relevant results on the differences in energy balance on different surfaces relating to the fraction of green vegetation on the Tibetan Plateau. However, I recommend some adjusts to be accepted for publication:
I suggest changing the title of the manuscript considering that the analysis of the green vegetation fraction was very important for the results and discussion of the work.
Line 60-111 – Introduction paragraph too long, I suggest splitting it into more paragraphs in order to make the reading simpler and more understandable.
Line 117-126 – I suggest removing or rewriting these lines. This text looks like methodology. It would be interesting to clarify the objective of the article.
Line 163-164 – Explain why you changed the spatial resolution of ERA5-Land from 0.1° x 0.1° to 0.25° x 0.25°?
Figure 8. Explain what the dashed line on the graph means.
Author Response
Response to Reviewer 3 Comments
The manuscript has relevant results on the differences in energy balance on different surfaces relating to the fraction of green vegetation on the Tibetan Plateau. However, I recommend some adjusts to be accepted for publication:
Point 1: I suggest changing the title of the manuscript considering that the analysis of the green vegetation fraction was very important for the results and discussion of the work.
Response 1: Thank you for this suggestion. The title of the manuscript has been changed to “Analysis of surface energy changes over different underlying surfaces based on MODIS land use data and green vegetation fraction over the Tibetan Plateau.”
Point 2: Line 60-111 – Introduction paragraph too long, I suggest splitting it into more paragraphs in order to make the reading simpler and more understandable.
Response 2: Thank you for raising this point. The introduction has been divided into five sections, which introduce the research background of the TP, the research status of land-atmosphere interaction, the latest achievements in surface energy and water research on the TP, the shortcomings and limitations of the research on the TP, and the main research contents of this paper. Please refer to the Introduction section of the manuscript (pages 1 to 3, lines 42–126).
Point 3: Line 117-126 – I suggest removing or rewriting these lines. This text looks like methodology. It would be interesting to clarify the objective of the article.
Response 3: Thank you for this suggestion. To clarify the objective of the paper, we have rewritten this paragraph to now read: “In this study, we used Normalized Difference Vegetation Index (NDVI) data from MODIS, ERA-5 Land reanalysis data, and long-term flux observation station data from six sites (BJ, MAWORS, NADORS, NAMORS, QOMS, and SETORS) in the TP of the Second Tibetan Plateau Scientific Expedition and Research to discuss the long-time series variation characteristics and energy distribution differences of the surface energy fluxes on different underlying surfaces over the TP” (page 3, lines 121–126).
Point 4: Line 163-164 – Explain why you changed the spatial resolution of ERA5-Land from 0.1° x 0.1° to 0.25° x 0.25°?
Response 4: Thank you for your comment. We acknowledge this error; the ERA5-Land data is indeed 0.1° × 0.1°. This error has been corrected accordingly in the manuscript. Please refer to page 3, line 164.
Point 5: Figure 8. Explain what the dashed line on the graph means.
Response 5: We thank the reviewer for raising this point. The dashed line indicates H=LE and H/LE=1. We have duly explained this in the respective figure. Please refer to page 13, line 386.

This manuscript is a resubmission of an earlier submission. The following is a list of the peer review reports and author responses from that submission.